# Evaluation of Biochemical Serum Markers for the Diagnosis of Polycystic Ovary Syndrome (PCOS) in Obese Women in Kazakhstan: Is Anti-Müllerian Hormone a Potential Marker?

**DOI:** 10.3390/biomedicines12102333

**Published:** 2024-10-14

**Authors:** Malika Madikyzy, Aigul Durmanova, Alexander Trofimov, Burkitkan Akbay, Tursonjan Tokay

**Affiliations:** 1Department of Internal Medicine, University Medical Center, 46 Syganak St., Astana 010000, Kazakhstan; malika.madikyzy@nu.edu.kz (M.M.); aiguldurmanova@mail.ru (A.D.); 2Department of Biology, School of Sciences and Humanities, Nazarbayev University, 53 Kabanbay Batyr Ave., Astana 010000, Kazakhstan; alexander.n.trofimov@gmail.com (A.T.); burkitkan.akbay@nu.edu.kz (B.A.)

**Keywords:** polycystic ovary syndrome (PCOS), obesity, biomarker, anti-Müllerian hormone (AMH), hyperandrogenism

## Abstract

**Background:** Polycystic Ovarian Syndrome (PCOS) is a common endocrine condition that affects 8–13% of women of reproductive age. In Kazakhstan, the prevalence of this syndrome is particularly high compared with other countries and the global average. Currently, the diagnosis of PCOS is based on internationally established Rotterdam criteria, using hyperandrogenism as a key parameter. These criteria are applied to diagnose PCOS in all female patients, although obese patients may have excess testosterone produced by adipose tissue. To avoid possible misdiagnosis, an additional criterion, especially for the diagnosis of PCOS in obese women, could be considered. The **aim** of this study was to identify whether anti-Müllerian hormone (AMH) or other biochemical criteria can be used for this purpose. **Methods:** A total of 138 women were recruited for this study and grouped into control (n = 46), obese subjects without PCOS (n = 67), and obese patients with PCOS (n = 25). The health status, anthropometric parameters, and serum indicators for glucose, glycosylated hemoglobin, and hormone levels were examined for all subjects. Statistical data were analyzed using GraphPad Prism 10 software for interpretation of the data. **Results:** Serum AMH, testosterone, and LH were positively correlated in obese PCOS patients, while AMH and FSH were negatively correlated. Compared with other biochemical indicators, the serum AMH and testosterone levels in obese PCOS patients were significantly higher than those in non-PCOS patients (regardless of obesity), and AMH was also positively correlated with testosterone. **Conclusions:** AMH appears to be a reliable criterion in addition to testosterone for the diagnosis of PCOS in obese women.

## 1. Introduction

There are many studies indicating the negative effects of obesity on the reproductive health of women. Women with obesity are more likely to suffer from reproductive system disorders, including menstrual dysfunction, anovulation, infertility, difficulties in conception, and miscarriage [1]. Polycystic ovary syndrome (PCOS) is one such pathology, and its prevalence worldwide was estimated at 276.35 per 100,000 women, with the number for Kazakhstan being significantly higher than worldwide and for many other countries—417.23 per 10,000 women [2]. The increased risk of PCOS is of particular importance because 38–88% of women with PCOS suffer from overweight or obesity [3]. Furthermore, PCOS is associated with metabolic disruptions, an increased incidence of complications during pregnancy, infertility, endometrial carcinoma, diabetes mellitus, glucose intolerance, dyslipidemia, and cardiovascular disorders in the long run [4,5]. Therefore, it is crucial to determine an indicator that would allow the identification of women with obesity who are at a higher risk of developing PCOS. If such an indicator is identified, it would be possible to prevent the development of PCOS and start treatment as early as possible. 

PCOS is characterized by a range of hormonal disruptions, including decreased level of follicle-stimulating hormone (FSH) and increased level of luteinizing hormone (LH), abnormal gonadotropin-releasing hormone (GnRH) secretion, elevated serum androgens and insulin [6]. There are numerous symptoms and signs that make the process of PCOS diagnosis complicated. For example, oligo-anovulation, polycystic ovaries found by ultrasound investigation and hyperandrogenism have to be considered to diagnose PCOS currently [6]. Due to the large number of these indicators, such identification can be complicated. 

The level of anti-Müllerian hormone (AMH) is an indicator that has a high potential to facilitate this process. AMH is a peptide hormone that is synthesized in granulosa cells from an inert AMH precursor (proAMH) that can be further cleaved to yield a COOH-terminal dimer (AMH_C_; 25 kDa) and NH2-terminal dimer (AMH_N_; 120 kDa), which remain associated in a noncovalent complex form (AMH_N,C_) [7]. Both proAMH and complex AMH_N,C_ can be present in the serum. AMH synthesis in granulosa cells begins with folliculogenesis and lasts until a woman undergoes menopause [8]. Similar to other TGF-β family members, AMH_C_ induces the SMAD pathway in order to regulate stages of folliculogenesis and various neuroendocrine processes [9]. Hence, measuring AMH at any time of the menstrual cycle will give the complete clinical information necessary to assess the ovarian reserve [10]. Disruptions of the ovarian reserve caused by diseases of the female endocrine system will affect AMH levels. Particularly, many studies show that if a woman suffers from PCOS, the level of AMH in the blood will be 2–3 times more than the normal level [11,12]. For this reason, the level of AMH can potentially reflect whether a woman with obesity is likely to develop PCOS. 

One of the Rotterdam criteria used to diagnose PCOS is hyperandrogenism or excessive testosterone level. According to Wagner et al., the ovaries and adrenal glands of women with obesity increase androgen production due to insulin resistance and hyperinsulinism. Moreover, Dr Wagner and her colleagues state that adipose tissue is the source of excess androgens in obese women because the androgen level in the cohort with obesity was significantly higher than in the lean cohort [13]. Among premenopausal women, adipose tissue is responsible for up to 50% of testosterone in serum [14]. This makes hyperandrogenism alone insufficient as a criterion to diagnose PCOS among obese women because not only PCOS but also obesity increases testosterone levels. It may confuse doctors and allow incorrect PCOS diagnosis and prescription of wrong treatment. 

Although there are many studies indicating a higher risk of PCOS development in women with obesity, few studies focus on how to facilitate the process of PCOS diagnosis and identification of individuals who are more predisposed to develop PCOS. The necessity of such studies is highlighted by Malhorta et al. They conclude that elevated AMH levels correspond to more alarming metabolic, endocrinological, and clinical criteria, thus making AMH useful in making treatment more personalized along with predicting responses to treatment and its outcomes [15]. Similarly, the study which focused on the relationship between levels of AMH and other metabolic/hormonal substances in women with PCOS from Bangladesh noted the need for further studies. It states that more data are necessary to evaluate AMH potential [16]. Moreover, the effectiveness of AMH in the differentiation of PCOS phenotypes was indicated in the study by Ngyuen et al. [17] The potential of AMH in the differentiation of PCOS phenotypes shows that it can be potentially useful in assessing the severity of the disease and making a prognosis. Therefore, the goal of this study is to identify if AMH or other biochemical criteria can be used as additional PCOS diagnostic criteria among obese women. Achieving this goal will help to make PCOS diagnosis more effective among patients with obesity to offer better treatment as early as possible. Moreover, AMH levels could be used to identify if a woman is at risk of PCOS development, thus allowing for offering her treatment and prevention therapies before PCOS develops. 

## 2. Materials and Methods

### 2.1. Selection of Subjects

This study involved 138 women who received medical consultations in the Department of Endocrinology at the Republican Diagnostic Center in the Republic of Kazakhstan. The inclusion and exclusion criteria of the sample selection are demonstrated in the flow chart (Figure 1). Briefly, women aged between 18 and 35 years were included in the study, while women with clinically significant oncologic or cardiovascular diseases, hypertensive disease of stages 2–3, congenital disorders of lipid metabolism, pregnancy, lactation, smoking habits, and menopause were excluded from the study. One group of non-obese participants without PCOS was recruited as a control group (n = 46). The obese women’s body mass index (BMI) exceeded 30 kg/m^2^. They were divided into two groups based on the diagnosis of PCOS: a total of 25 of them were designated to the group with PCOS (n = 25), and the remaining 67 were designated to the group without PCOS (n = 67) based on their diagnosis.

PCOS diagnosis was accomplished based on the Rotterdam Consensus of the European Experts. A woman is diagnosed with PCOS if she has any two of the following symptoms: (1) irregular menses (IM) (≤6 per year), oligoovulation, and/or anovulation; (2) excess androgen activity and secretion (hirsutism, acne) and/or biochemical hyperandrogenism; and (3) typical changes on gynecologic ultrasound (≥12 follicles sized 2–9 mm).

This study was conducted in accordance with the Declaration of Helsinki and approved by the 46th Ethics Committee of the Republican Diagnostic Center (approval No. 0215PK02290). Informed consent was obtained from all subjects involved in this study. Written informed consent has been obtained from the patients to publish this paper.

### 2.2. Intervention of Subjects

Further investigation was based on the data obtained via the methods described below.

(1)Survey of patients: this was conducted to collect anamnestic data to assess health and heredity.(2)Anthropometric data: Measurements included height, weight, waist circumference, hip circumference, and blood pressure. The body mass index (BMI) was calculated using Quetelet’s index formula: weight (kg)/height (m)2. Obesity was considered abdominal when the ratio of waist -to-hip measurement (WM/HM) in women was more than 0.8.

### 2.3. Biochemical Analysis

Blood samples were obtained between 08:00 and 10:00 a.m. from women who were on days 7 to 11 of their menstrual cycle and had fasted for at least 12 h. The blood was collected three times for each subject within six months. The samples were centrifuged at 2000× *g* for 10 min. Serum samples were stored at −23 °C until measurements were performed. 

(1)Lipid profile: Total cholesterol and triglycerides were measured by the enzymatic colorimetric method using a COBAS INTEGRA analyzer (Roche Diagnostics, Basel, Switzerland). Low-density lipoproteins (LDL) and high-density lipoproteins (HDL) were measured by the homogeneous enzymatic colorimetric method using a Roche/Hitachi cobas c analyzer (Roche Diagnostics, Basel, Switzerland).(2)Fasting glucose level: measured by the standard enzymatic method using a Roche/Hitachi cobas c analyzer (Roche Diagnostics, Basel, Switzerland).(3)Glycated hemoglobin (HbA1c): measured by a turbidimetric inhibition immunoassay using a Roche analyzer (Roche Diagnostics, Basel, Switzerland).(4)Hormone levels: Levels of thyrotropic hormone, prolactin, estrogens, progesterone, follicle-stimulating hormone (FSH), luteinizing hormone (LH), testosterone, and insulin were measured by Eclia electrochemiluminescence immunoassay using a Elecsys and cobas e analyzer (Beckman Coulter, Inc., Brea, CA, USA). The level of anti-Müllerian hormone (AMH) was measured by ELISA. Levels of adiponectin and resistin were measured by the ELISA method using specific monoclonal antibodies.

### 2.4. Ultrasonograph

Ultrasound investigations (U/S): these were carried out on the seventh day of the menstrual cycle using a Logiq E9 (GE HealthCare, Chicago, IL, USA) with ML6-15 Mhz linear array transducer and IC 5-9 Mhz transvaginal curvilinear transducer. The investigations included 250 pelvic ultrasounds, 250 thyroid ultrasounds, and 250 breast ultrasounds.

### 2.5. Statistical Analysis

The power calculation was performed using G*Power 3.1.9.7 software. With an effect size of 0.8 and a sample size of 25 participants in the PCOS group, the power was 0.92 for the group comparison analysis and 0.99 for the correlational analysis. Statistical data were analyzed using GraphPad Prism 10 software (GraphPad Software, San Diego, CA, USA) and Microsoft Excel 365 (Microsoft Corporation, Redmond, WA, USA). The normality of the data distribution was tested using the D’Agostino–Pearson criterion. The Kruskal–Wallis nonparametric test and post hoc Dunn’s test were used to compare median values of the indicators among the control group and obese women with and without PCOS. The median values and inter-quartile ranges (IQR) are indicated in the text. Descriptive statistical data, as well as the results of the Kruskal–Wallis nonparametric test and post hoc Dunn’s test, are summarized in Appendix A. Two-tailed Spearman’s correlation analysis was used to relate the level of AMH in the blood to other hormonal parameters (for the AMH–Testosterone correlation analysis in obese PCOS patients, a one-tailed Spearman’s test was used, as we hypothesized a positive correlation based on data suggesting that in PCOS, elevated testosterone stimulates the production of AMH [18]). ROC analysis was used to test the specificity and sensitivity of AMH as a biomarker for PCOS diagnosis. The area under the curve (AUC) was calculated in GraphPad Prism to assess how well AMH distinguishes women with and without PCOS, where an AUC of 0.5 indicates no discrimination and 1.0 indicates perfect discrimination. A *p*-value of less than 0.05 was considered statistically significant.

## 3. Results

### 3.1. General Parameters

A total of 138 female subjects aged 36 years or younger were included in this study. The age and BMI parameters of subjects are demonstrated in Figure 2 and Appendix A. The median ages of the obese women with and without PCOS and control subjects are 28.00 (24.75; 30.25), 30.00 (26.00; 34.00), and 30.00 (26.00; 33.00), respectively. There is no statistically significant difference in age between each group (*p* > 0.05; Figure 2A). In terms of BMI, there is no statistical difference in BMI between obese women with and without PCOS (33.50 (31.38; 36.03) and 34.25 (31.05; 38.48) kg/m^2^, respectively; *p* > 0.9999). However, there is a significant difference between the control (21.40 (19.73; 23.63) kg/m^2^) and test subjects (i.e., obese women with PCOS and without PCOS women, both *p* < 0.0001).

### 3.2. Hormonal Parameters

Serum levels of hormones in control and study groups (obese group with or without PCOS are shown in Figure 3 and Appendix A. The median level of AMH among obese women with PCOS (9.67 (5.89; 12.65) ng/mL) is higher than in no PCOS group (1.70 (0.72; 2.64) ng/mL; *p* < 0.0001) and control group (4.00 (2.62; 6.86) ng/mL; *p* = 0.0232). Moreover, the AMH level of the control group is significantly higher than this parameter in the no PCOS group (*p* < 0.0001).

The second parameter with some statistically significant differences between the groups is testosterone. Serum concentrations of this parameter are significantly higher in obese subjects with PCOS (1.42 (1.09; 1.61) ng/mL) than in obese subjects without PCOS (0.21 (0.14; 0.30) ng/mL; *p* < 0.0001; Figure 3B). Likewise, obese women with PCOS have higher median testosterone than the control group (0.19 (0.10; 0.30) ng/mL; *p* < 0.0001). However, no significant difference was found between obese women who were not diagnosed with PCOS and the control group (*p* > 0.9999).

In stark contrast, the FSH activity level is significantly lower in the PCOS group (4.44 (3.48; 5.72) mIU/mL) than in no PCOS (5.82 (4.73; 7.08) mIU/mL; *p* = 0.0050) and control (6.49 (5.60; 7.50) mIU/mL; *p* = 0.0003; Figure 3C) groups. The latter two groups were not statistically different in median FSH activity level (*p* = 0.7428).

The next parameter with statistically significant differences between the groups is LH activity (Figure 3D). Serum levels of LH activity for women with obesity and PCOS (9.73 (8.55; 10.79) mIU/mL) are significantly larger than those of obese women without PCOS (4.50 (3.35; 7.30) mIU/mL; *p* < 0.0001) and subjects from the control group (5.88 (4.45; 7.08) mIU/mL; *p* = 0.0007). Despite this, the no PCOS and control groups did not significantly differ (*p* = 0.5655). 

Although most other parameters above differ between obese subjects with PCOS and other groups, estradiol levels show no significant differences between the PCOS and no PCOS groups (38.83 (23.00; 82.53) pg/mL and 31.46 (19.87; 50.54) pg/mL, respectively; *p* = 0.4739; Figure 3F) or between PCOS and control groups (47.14 (31.08; 74.52) pg/mL; *p* = 0.7826). On the other hand, median estradiol levels are significantly lower in the no PCOS group compared with control subjects (*p* = 0.0043). No significant findings could be observed in TSH (Figure 3E), progesterone (Figure 3G), and prolactin levels (Figure 3H).

There were more statistically significant levels in terms of insulin. Median serum levels of insulin activity among subjects with PCOS and obesity (15.90 (11.34; 26.04) uIU/mL) are higher than among control subjects (7.10 (5.52; 9.69) uIU/mL; *p* < 0.0001; Figure 3I). Obese women without PCOS (17.62 (13.15; 25.05) uIU/mL; *p* < 0.0001) also have higher insulin levels than the control group. On the contrary, no statistically significant differences in insulin levels were observed between obese subjects with and without PCOS (*p* > 0.9999). 

The opposite trend was found in adiponectin levels. PCOS and no PCOS groups had adiponectin levels (7.99 (6.41; 12.25) ng/mL and *p* < 0.0001; 8.66 (5.58; 12.67) ng/mL and *p* < 0.0001; respective order; Figure 3J) lower than control group (16.15 (13.53; 19.40) ng/mL), even though they did not significantly differ from one another (*p* > 0.9999). Another peculiar trend can be observed in resistin levels. This parameter among obese subjects with PCOS (8.09 (3.61; 9.86) ng/mL; *p* = 0.0042; Figure 3K) and without PCOS (6.54 (4.70; 11.98) ng/mL; *p* < 0.0001) was higher than among control subjects (3.74 (1.80; 5.56) ng/mL). However, the two groups with obesity did not differ significantly (*p* = 0.7897).

The AMH level in the blood was further analyzed for its specificity and sensitivity for being used as a biomarker for PCOS diagnosis and was found to be highly reliable for this purpose (area under the curve = 0.9069; *p* < 0.0001; Figure 4). At the AMH threshold of 8.173 ng/mL, the sensitivity equals 90.00% (95% CI: 69.90% to 98.22%), and specificity is 97.26% (95% CI: 90.55% to 99.51%), indicating that at this level, 90% of women with PCOS can be correctly identified, while 97.26% of women without PCOS would be correctly classified if their AMH level is below 8.173 ng/mL. The likelihood ratio is 32.85, showing that women with AMH above this level are 32 times more likely to have PCOS.

### 3.3. Metabolic Parameters

Serum levels of metabolites in control and study groups (obese group with or without PCOS are shown in Figure 5 and Appendix A. Glucose median levels among obese women with PCOS (5.23 (4.99; 5.84) mmol/L; *p* = 0.0003; Figure 5A) and without PCOS (5.31 (4.92; 5.74) mmol/L; *p* < 0.0001) were higher than in the control group (4.79 (4.59; 5.16) mmol/L), even though there was no statistically significant difference between the two groups with obesity (*p* > 0.9999). Despite significant findings in terms of glucose levels, the level of glycated hemoglobin is not different between the PCOS (4.30 (3.62; 5.53) %), no PCOS (4.08 (3.81; 5.51) %), and control (4.54 (3.70; 5.01) %) groups (*p* > 0.05; Figure 5B). 

Regarding cholesterol serum concentration, the only statistically significant difference is found between obese women who were not diagnosed without PCOS (4.89 (4.42; 6.29) mmol/L) and the control group (4.19 (3.83; 4.91) mmol/L; *p* < 0.0001; Figure 5C). Subjects with obesity and PCOS (4.58 (4.09; 5.32) mmol/L) did not significantly differ from the no PCOS (*p* = 0.1699) and control groups (*p* = 0.3262).

Another statistically significant difference exists in HDL levels between obese subjects with PCOS and control subjects (1.07 (0.89; 1.23) mmol/L and 1.50 (1.33; 1.70) mmol/L, respectively; *p* < 0.0001; Figure 5D). Similarly, the HDL level among obese subjects without PCOS (1.21 (1.01; 1.45) mmol/L) is lower than in the control group (*p* = 0.0003). However, the groups with obesity did not show statistically significant differences (*p* = 0.1666). 

LDL is one more metabolic parameter that showed statistically significant results. Both obese patients with and without PCOS (3.02 (2.44; 3.49) mmol/L and *p* < 0.0001; 3.12 (2.67; 3.49) mmol/L and *p* < 0.0001, the order is respective; Figure 5E) had LDL levels which were higher than in the control group (1.98 (1.71; 2.25) mmol/L), but the two groups with obesity were not statistically different (*p* > 0.9999).

A trend similar to LDL was observed in terms of triacylglycerol. The median levels of this parameter are higher in obese women with and without PCOS (1.57 (1.07; 2.17) mmol/L and *p* < 0.0001; 1.55 (0.90; 2.53) mmol/L and *p* < 0.0001, the order is respective; Figure 5F) compared with the control subjects (0.60 (0.51; 0.81) mmol/L). Triacylglycerol levels of obese subjects suffering from PCOS are not significantly different from those of obese patients without PCOS (*p* > 0.9999).

### 3.4. Correlation between AMH and Some Hormonal Parameters

A statistically significant negative correlation can be observed between AMH and TSH in the control group (r = −0.29; *p* < 0.05; Figure 6). Another significant negative correlation can be noted between AMH and FSH (r = −0.32; *p* < 0.01) among obese women without PCOS.

As for positive correlations, one can be noted between AMH and adiponectin (r = 0.24; *p* < 0.05) among overweight women without PCOS. Similarly, there was a positive correlation between AMH and progesterone (r = 0.35; *p* < 0.01) in the same group. One more positive correlation was noticed between AMH and testosterone (r = 0.38; *p* < 0.05) for obese women suffering from PCOS.

## 4. Discussion

The Rotterdam criteria include hyperandrogenism as one of the factors used to differentiate women with PCOS from those without PCOS. Secretion of testosterone by excessive adipose tissue may hinder its reliability as an independent diagnostic criterion. Moreover, obesity induces considerable changes in the interplay between different metabolic and hormonal parameters. Therefore, there is a need to introduce additional criteria for PCOS diagnosis among obese women. There is a significant correlation between AMH and testosterone among obese women suffering from PCOS. Furthermore, changes in normal interplay between hormonal parameters induced by obesity are represented by correlations between AMH and other hormonal parameters such as TSH, adiponectin, progesterone, and FSH. According to Lv et al., women suffering from PCOS at a young age exhibit symptoms such as high testosterone levels and chronic anovulation, whereas older women primarily experience disruptions in metabolic parameters, obesity, and insulin resistance [19]. The results of our study show that obese women with PCOS are significantly younger than obese women without PCOS. Additionally, there are differences in metabolic parameters between these two groups, including cholesterol and HDL levels. Using AMH as a criterion in addition to testosterone would help to differentiate changes in the hormonal status of patients caused by obesity along with age and changes induced by PCOS. This hypothesis is supported by several studies that highlight a significant correlation of AMH with various PCOS features, including morphology, oligo/amenorrhea, and elevated testosterone levels [20,21,22]. 

One of the possible explanations for this could be disruptions in the regulation of AMH expression, which is mediated by testosterone. Dilaver et al. (2019) suggested that testosterone is converted to estradiol by aromatase [18]. Estradiol affects ERβ receptors, thus reducing AMH levels. In addition, a precursor of testosterone called dihydrotestosterone directly impedes pathways of AMH synthesis. In patients with PCOS, increased expression of Erα receptors alters the receptor ratio necessary for this mechanism and elevates AMH level. Furthermore, extremely high levels of dihydrotestosterone often discovered among PCOS patients induce a dose-related increase in AMH synthesis. The results of their in vitro experiments confirmed that these interactions indeed could take place, even though more studies are needed to make this model more reliable. The results of our study support this finding because a positive correlation was observed between AMH and testosterone in the PCOS group, while there is no significant correlation in other groups.

The results of many studies suggest that PCOS does not affect the correlation between AMH and FSH. High FSH levels inhibit AMH secretion in infantile mice and cows [23,24]. The results of our study suggest that there is no negative correlation between FSH and AMH in obese women suffering from PCOS and a healthy control group, as the Spearman coefficients are not significant. Meanwhile, a significant negative correlation was found between AMH and FSH for obese women without PCOS.

To explain this correlation, numerous studies investigated the interaction between AMH and FSH. One of the studies states that factors GDF9 and BMP15 from oocytes are involved in the regulation of AMH in granulosa cells synthesizing it. In particular, these factors stimulate p300-mediated acetylation of histones in nuclei of granulosa cells by cooperating with Smad2/3 and PI3K/Akt signaling. As a result, the genes involved in AMH synthesis are expressed because access to DNA becomes easier. Repression of this pathway is realized by Gonadotropin-Inducible Ovarian Transcription Factor 1 (GIOT1) belonging to a family of zinc-finger proteins. GIOT1 has a Krüppel-associated box A domain that functions as a repressor of transcription, which can directly interact with either/both Smad2/3 and p300. Moreover, GIOT1 expands another protein with repressor functions called HDAC2. This mechanism also mediates the inhibition of AMH expression by FSH [25]. The inhibition typically results in a negative correlation between these two hormones. 

Our findings partially correspond with this mechanism because there is a significant negative correlation between AMH and FSH among obese women without PCOS. The absence of this correlation in the PCOS group could be due to the hormonal disbalance observed in these patients. Such disbalance could lead to the prevalence of other mechanisms of AMH regulation. Therefore, further studies are needed to investigate how PCOS disrupts FSH-mediated inhibition of AMH. One more discrepancy is that no significant correlation of these substances was found for healthy control subjects. This highlights the need for further investigations of other hormonal mechanisms involved in AMH regulation. Despite these discrepancies, AMH remains consistently high in patients suffering from PCOS. Hence, it can be concluded that AMH is a useful marker for diagnosing PCOS. Obesity can disrupt either the expression or the effect of GDF9 and BMP15 factors, leading to the absence of a correlation between AMH and FSH.

Estradiol levels are usually low in women suffering from PCOS. In a study by Masjedi and colleagues, a cohort of women with PCOS had significantly lower estradiol levels compared with healthy women [26]. Among women with obesity or excess weight, estradiol levels were found to be lower than in women without obesity, regardless of other factors like age, smoking, and race [14]. Although the mechanism explaining the relationship between low estradiol and obesity is not fully understood, it is clear that disruptions in estradiol metabolism are involved in the causation of obesity in women. AMH-mediated inhibition of aromatase enzyme that converts testosterone to estradiol can be one of the possible explanations, as it was described in the previous paragraph. The results of this study partly align with expectations, as female obese patients without PCOS had significantly lower estradiol levels than normal-weight women. The unexpected finding was that women with both PCOS and obesity had estradiol levels almost the same as the control cohort. Even though the mechanism underlying this observation is unclear, the existence of an interplay between obesity and PCOS is supported by this finding. It also helps to address the goal of this study, as such inconsistency makes estradiol an inappropriate indicator of PCOS among obese women.

Resistin and adiponectin are two important adipokines secreted by the adipose tissue. They act like hormones and play an important role in regulating metabolic activity. Numerous studies showed a possible association of resistin and adiponectin with obesity and PCOS. In our study, serum levels of resistin were found to be significantly increased in the obese subjects (*p* < 0.0001) and in PCOS patients (*p* < 0.01) when compared with controls, whereas adiponectin was found to be significantly lower (*p* < 0.0001) in both obesity and PCOS. There was no significant difference in both adiponectin and resistin levels between obese and PCOS groups. At least two studies support our findings. Seow et al. (2004) reported an over-expression of resistin in obesity and PCOS patients and suggested that resistin could be a contributing factor to the pathogenesis of PCOS [27]. Hu et al. (1996) showed that the expression of adiponectin mRNA was significantly reduced in obesity [28]. To address the possible association between these two adipokines and PCOS, Nambiar et al. (2016) analyzed the genetic polymorphisms of adiponectin and resistin and found that serum adiponectin and resistin levels were significantly correlated with BMI but not with PCOS [29]. 

The positive correlation between AMH and adiponectin, as well as the reduced level of AMH in women with obesity without PCOS, suggests that excessive adipose tissue typically suppresses AMH secretion. Some previous research reports that obesity may negatively impact AMH levels [30]. However, because there is no such correlation in patients with both PCOS and obesity, and they exhibit excessive AMH levels, it appears that the suppression of AMH by adipose tissue can be disrupted by PCOS. This further supports the hypothesis that normal ranges of biochemical indicators cannot be used to diagnose reproductive diseases in women with obesity. Obesity may exacerbate disruptions caused by reproductive diseases to such an extent that normal ranges of biochemical indicators do not show deviations characteristic of a particular disease.

Another biochemical criterion affected by PCOS is progesterone level. Numerous studies report that progesterone levels are decreased among women who suffer from PCOS. Research conducted by Estienne and his colleagues reveals that women of normal weight who were diagnosed with PCOS had lower progesterone levels than women without PCOS [31]. Also, the study by Daghestani and his colleagues found that a cohort diagnosed with PCOS had progesterone levels approximately five times lower than those in the control group [32]. Surprisingly, in this research, there was no significant difference in progesterone levels between women with obesity and PCOS and the healthy control group. In contrast, women with obesity without PCOS had lower progesterone levels. This finding indicates that progesterone is affected by both obesity and PCOS, making it inappropriate as an indicator of PCOS among obese women. Furthermore, there is a significant positive correlation between AMH and progesterone among women with obesity without PCOS. Because this tendency is not observed in both control and PCOS groups, it is likely that hormonal disruptions induced by obesity alone can interfere with the normal interplay between AMH and progesterone metabolic pathways. 

As to the negative correlation between AMH and TSH in the healthy control group, this finding coincides with previous studies. A decrease in levels of TSH, along with a decline in the number of antral follicles produced, was observed in one of the studies [33]. Because antral and pre-antral follicles secrete AMH [34], it is rational to expect that healthy subjects will have an inverse relationship between these two substances.

Regarding LH, numerous studies report elevated blood levels of this hormone in women who suffer from PCOS. According to a study conducted by Ambiger and colleagues, LH levels were elevated among women diagnosed with PCOS compared to healthy women [35]. One possible mechanism behind this phenomenon could be constant disruptions in estrogen and progesterone levels, which impair the function of the hypothalamic pulse generator. This leads to the pituitary gland becoming excessively sensitive to gonadotropin-releasing hormone (GnRH) or changes in the secretion patterns of GnRH, resulting in elevated LH levels [36]. Similar findings are reported in the study by Shrivastava and Conigliaro, which states that hyperinsulinemia partially leads to PCOS by elevating LH and GnRH levels through its influence on both the pituitary gland and hypothalamus [37]. 

The results of our study confirm elevations in LH blood concentration. However, it is interesting to note that many obese women who do not suffer from PCOS also have elevated LH concentrations, suggesting that obesity disrupts the regulation of LH secretion, which may lead to symptoms resembling PCOS in obese women. However, there were no statistically significant differences in blood LH levels between obese women with PCOS and those without PCOS. Therefore, LH levels cannot be used as a criterion to diagnose PCOS.

## 5. Conclusions

Overall, the results of this study supported the hypothesis proposed at the beginning of the study. The other indicators, except AMH and testosterone, were mostly inconsistent with what was expected in women with PCOS or obesity, indicating that obesity and PCOS interfere with normal metabolism and disrupt biochemical indicators. Such extreme disruption of indicators complicates the diagnosis of reproductive system diseases, including PCOS. Therefore, there is a need to introduce additional biochemical criteria to accurately and consistently diagnose reproductive disorders in obese patients. For PCOS in particular, AMH showed consistency because it was only higher in women with obesity and PCOS. Hence, AMH appears to be a more appropriate additional biochemical criterion for the diagnosis of PCOS compared with other biochemical indicators. However, more studies on the metabolic relationship between obesity and PCOS are needed to assess the relevance of AMH in this role. 

## Figures and Tables

**Figure 1 biomedicines-12-02333-f001:**
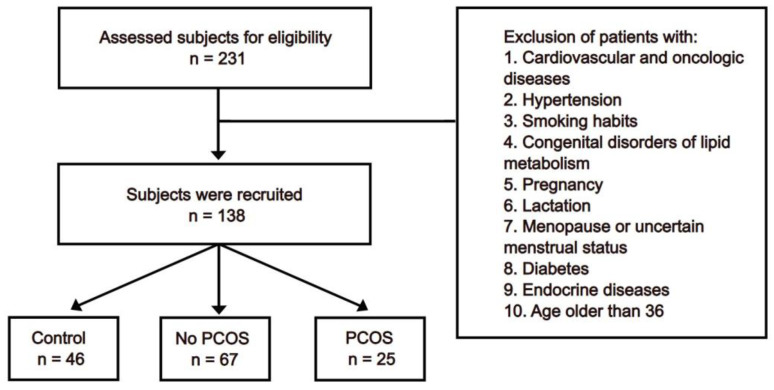
Flow chart representing the selection of subjects for the control group (n = 46), obese women without polycystic ovary syndrome (no PCOS: n = 67), and obese patients with polycystic ovary syndrome (PCOS: n = 25).

**Figure 2 biomedicines-12-02333-f002:**
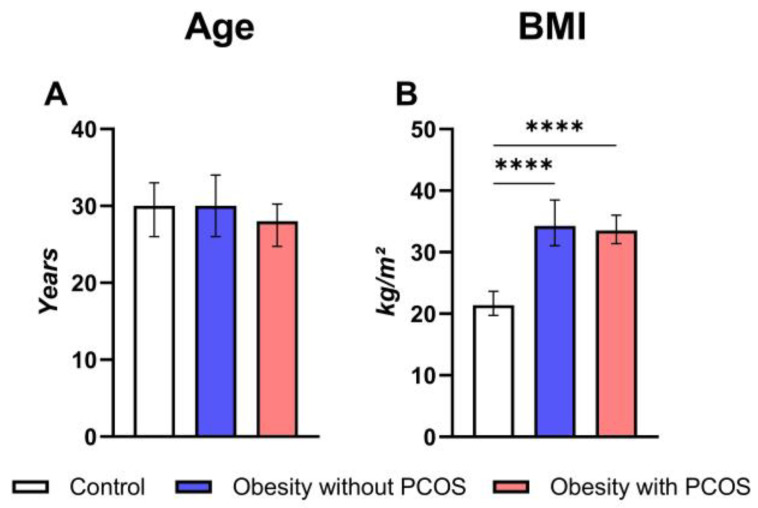
Comparison between control group (n = 46) and study groups (obese group with or without PCOS (PCOS: n = 25; no PCOS: n = 67)) in terms of age and body mass index (BMI). (**A**) Age; (**B**) BMI. Kruskal–Wallis nonparametric test and post hoc Dunn’s test. **** *p* < 0.0001. Median and IQR.

**Figure 3 biomedicines-12-02333-f003:**
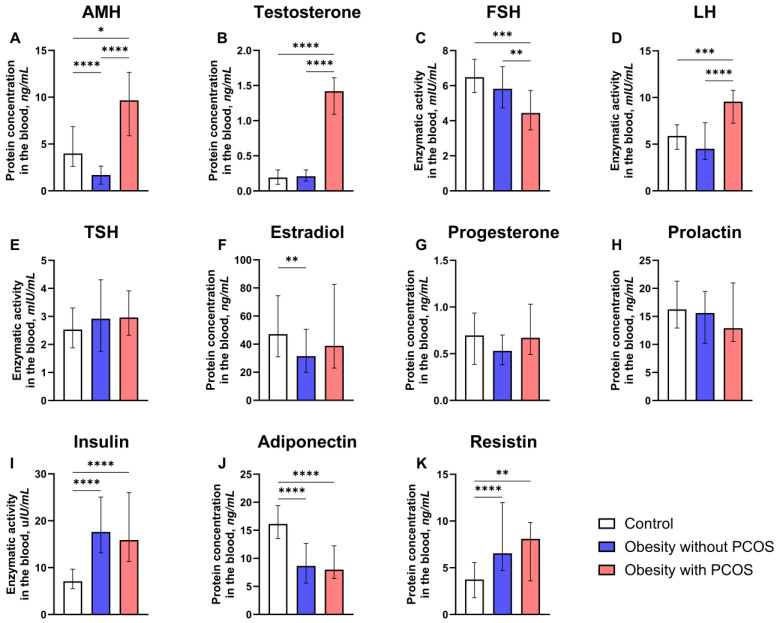
Comparison between control group (n = 46) and study groups (obese group with or without PCOS (PCOS: n = 25; no PCOS: n = 67) in terms of serum levels of hormones. (**A**) Serum anti-Müllerian hormone (AMH) protein concentration; (**B**) serum testosterone protein concentration; (**C**) serum follicle-stimulating hormone (FSH) level of enzymatic activity; (**D**) serum luteinizing hormone (LH) level of enzymatic activity; (**E**) serum thyroid-stimulating hormone (TSH) level of enzymatic activity; (**F**) serum estradiol protein concentration; (**G**) serum progesterone protein concentration; (**H**) serum prolactin protein concentration; (**I**) serum insulin level of enzymatic activity; (**J**) serum adiponectin protein concentration; (**K**) serum resistin protein concentration. Kruskal–Wallis nonparametric test and post hoc Dunn’s test. * *p* < 0.05, ** *p* < 0.01, *** *p* < 0.001, **** *p* < 0.0001. Median and IQR.

**Figure 4 biomedicines-12-02333-f004:**
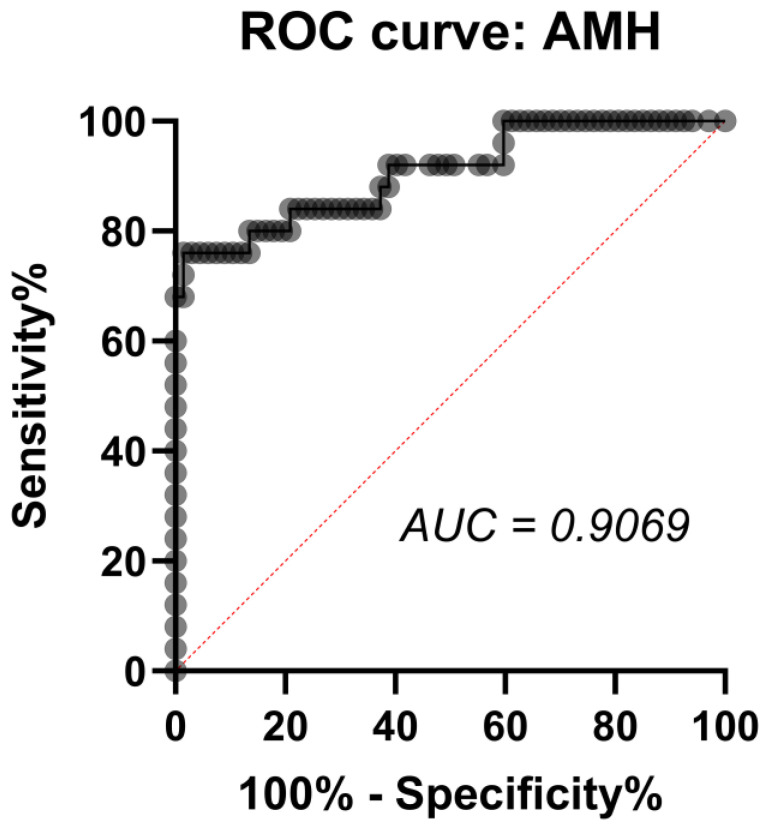
ROC analysis of AMH specificity and sensitivity as a biomarker for PCOS diagnosis. Black circles indicate the pair of [sensitivity%] and [100% − specificity%] for each value of AMH level in the blood serum of obese women with or without PCOS.

**Figure 5 biomedicines-12-02333-f005:**
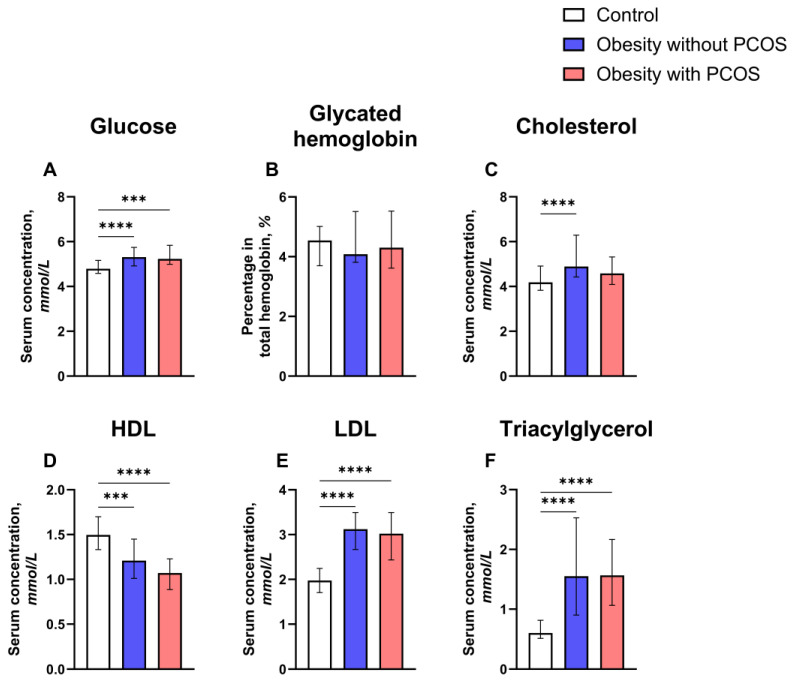
Comparison between control group (n = 46) and study groups (obese group with or without PCOS (PCOS: n = 25; no PCOS: n = 67)) in terms of serum metabolites. (**A**) Serum glucose level; (**B**) serum glycated hemoglobin level; (**C**) serum cholesterol level; (**D**) serum high-density lipoprotein (HDL) level; (**E**) serum low-density lipoprotein (LDL) level; (**F**) serum triacylglycerol level. Kruskal–Wallis nonparametric test and post hoc Dunn’s test. *** *p* < 0.001, **** *p* < 0.0001. Median and IQR.

**Figure 6 biomedicines-12-02333-f006:**
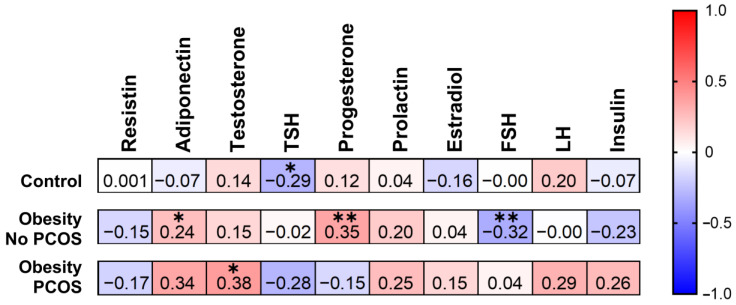
Correlation coefficients from Spearman correlation analysis of AMH and hormonal parameters including testosterone, FSH, LH, TSH, estradiol, progesterone, prolactin, and insulin for patients from the control group (no obesity, no PCOS; n = 46), obese patients without polycystic ovary syndrome (no PCOS; n = 67), and obese patients with polycystic ovary syndrome (PCOS; n = 25). * *p* < 0.05, ** *p* < 0.01. FSH, follicle-stimulating hormone; LH, luteinizing hormone; PCOS, polycystic ovary syndrome; TSH, thyroid-stimulating hormone.

## Data Availability

All data supporting the findings of this study can be made available to researchers upon request.

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
