# Peer review of "Evaluation of Biochemical Serum Markers for the Diagnosis of Polycystic Ovary Syndrome (PCOS) in Obese Women in Kazakhstan: Is Anti-Müllerian Hormone a Potential Marker?"

_biomedicines, 2024, doi:10.3390/biomedicines12102333_

Round 1

Reviewer 1 Report

Comments and Suggestions for Authors

Dear authors, I went through the manuscript entitled "Evaluation of biochemical serum indicators for polycystic ovarian syndrome diagnosis among women with obesity". The first question that came to my mind was about the novelty aspect of the present study. The risk factors of PCOS and its markers are well-studied. So overall, I have some concerns regarding the design and organisation of the manuscript. Please find my comments below;

1. The novelty element should be specified in the title and abstract itself. As I understand, this is one of the limited studies from Kazakhstan. Hence, the regional specificity of the study can be highlighted.

2. The abstract explained the background and methodology aspects of the study. However, major findings in terms of quantitative data should be included along with it. Hence, I suggest to re-write the abstract section, emphasizing findings and conclusion.

3. Methodology part can be written in a standard format. Authors can properly sub-divide it to different sections as sample selection (inclusion and exclusion criteria), Biochemical analysis, Ultrasonograph, Statistical analysis etc. This will make the manuscript more organized form

4. What was the sampling period of your study? How many months you took for data collection. Is the sample size of 138 enough to represent a population?

4. The result and discussion sections can be combined in the present manuscript. It enables proper analysis of the obtained results. At present, the results seems to be insignificant in most cases. So a combined Results & Discussion Session is appropriate.

5. There should be a defined conclusion section at the end.

Author Response

Reviewer 1

Dear authors, I went through the manuscript entitled "Evaluation of biochemical serum indicators for polycystic ovarian syndrome diagnosis among women with obesity". The first question that came to my mind was about the novelty aspect of the present study. The risk factors of PCOS and its markers are well-studied. So overall, I have some concerns regarding the design and organisation of the manuscript. Please find my comments below;

Thank you for the reviewer's insightful comments!

  1. The novelty element should be specified in the title and abstract itself. As I understand, this is one of the limited studies from Kazakhstan. Hence, the regional specificity of the study can be highlighted.

The novelty of the study is that AMH can be used as a diagnostic marker in addition to testosterone for the diagnosis of PCOS in obese women. This content has been added to the abstract and the title has been changed accordingly.

New title is: Evaluation of biochemical serum markers for the diagnosis of polycystic ovary syndrome (PCOS) in obese women in Astana, Kazakhstan: Is anti-Müllerian hormone a potential marker?

  1. The abstract explained the background and methodology aspects of the study. However, major findings in terms of quantitative data should be included along with it. Hence, I suggest to re-write the abstract section, emphasizing findings and conclusion.

The Abstract has been re-written to emphasize the findings and conclusions. Hopefully, it will now meet the required standards.

  1. Methodology part can be written in a standard format. Authors can properly sub-divide it to different sections as sample selection (inclusion and exclusion criteria), Biochemical analysis, Ultrasonograph, Statistical analysis etc. This will make the manuscript more organized form

We sincerely appreciate the reviewer’s pertinent comments!

The Materials and Methods section have been re-structured in a standard format as the reviewer’s recommendation.

  1. What was the sampling period of your study? How many months you took for data collection. Is the sample size of 138 enough to represent a population?

Thank you for the questions.

Blood samples were obtained between 08:00 and 10:00 a.m. after the women had fasted for at least 12 hours. The blood was collected three times for each subject within six months. The total duration of the sample collection was 16 months, Sep 2013 – Dec 2014.

According to many statisticians a sample size of 100 is the minimum you need for meaningful results. Therefore 138 is enough to represent a population. (Martínez-Mesa J et al. doi: 10.1590/abd1806-4841.20143705. PMID: 25054748).

  1. The result and discussion sections can be combined in the present manuscript. It enables proper analysis of the obtained results. At present, the results seem to be insignificant in most cases. So a combined Results & Discussion Session is appropriate.

Dear reviewer, we really appreciate your worthy comments and suggestions. However, in respecting the journal’s style, the Results section of the manuscript has been completely revised and re-written in a clearly arranged and logical order to make it easier to understand and analyze. Hopefully, you agree for this.

  1. There should be a defined conclusion section at the end.

We agree with the relevant reviewer's comment. Thank you!

A paragraph of conclusion is added to the manuscript.

Reviewer 2 Report

Comments and Suggestions for Authors

The manuscript under review addresses a crucial and timely topic of significant importance in PCOS and obesity research.

The methods and results presented in the manuscript are commendably clear and well-structured, a testament to the author's expertise in the field and making the content easily digestible for the reader.

The novelty of the approach is not appropriately stated and justified; even though the manuscript describes women with PCOS and obesity, these should be highlighted as core features, starting from the introduction to the conclusion.

The inclusion of highly relevant reference(s) will further support the objective approach, demonstrating how performed research builds upon and contributes to the existing body of knowledge in the field.

Author Response

Reviewer 2

1. The manuscript under review addresses a crucial and timely topic of significant importance in PCOS and obesity research.

We sincerely appreciate your comments. 

2. The methods and results presented in the manuscript are commendably clear and well-structured, a testament to the author's expertise in the field and making the content easily digestible for the reader.

Thank you.

3. The novelty of the approach is not appropriately stated and justified; even though the manuscript describes women with PCOS and obesity, these should be highlighted as core features, starting from the introduction to the conclusion.

The novelty of this study is that AMH can be used as a diagnostic marker in addition to testosterone for the diagnosis of PCOS in obese women. This content has been added to the abstract and in the title.

4. The inclusion of highly relevant reference(s) will further support the objective approach, demonstrating how performed research builds upon and contributes to the existing body of knowledge in the field.

We appreciate reviewer’s comment.

As you suggested, three recent and highly relevant review included in the manuscript.

  • Sivanandy, M.S.; Ha, S.K. The Role of Serum Anti-Mullerian Hormone Measurement in the Diagnosis of Polycystic Ovary Syndrome. Diagnostics (Basel) 2023, 13, doi:10.3390/diagnostics13050907.
  • Vural, F.; Vural, B.; KardaÅŸ, E.; Ertürk CoÅŸkun, A.D.; Yildirim, İ. The diagnostic performance of antimullerian hormone for polycystic ovarian syndrome and polycystic ovarian morphology. Arch. Gynecol. Obstet. 2023, 307, 1083–1090, doi:10.1007/s00404-022-06874-w.
  • Piltonen, T.T.; Allegranza, D.; Hund, M.; Buck, K.; Sillman, J.; Arffman, R.K. Validation of an Anti-Müllerian Hormone Cut-off for Polycystic Ovarian Morphology in the Diagnosis of Polycystic Ovary Syndrome in the HARMONIA Study: Protocol for a Prospective, Noninterventional Study.

Reviewer 3 Report

Comments and Suggestions for Authors

Dear Authors,

Thank you for your efforts in conducting this study. I found your work very interesting and would like to provide the following suggestions for improvement:

Please ensure the abstract is structured according to the appropriate study checklist.

The Methods and Results sections appear incomplete. Kindly revise and expand them to include all necessary details.

Materials and Methods:

Please clarify the type of study. Is it retrospective or prospective? Is it a cohort or case-control study?

How were the patients selected?

When was the study conducted? What was the sample size formula? Why were the specified numbers of women included in the different groups?

Were all lab tests and ultrasounds conducted by the same team or laboratory?

Statistical Analysis:

Please provide details about the ROC analysis and how the area under the curve (AUC) was calculated. This should be included in the statistical analysis section.

Outcome:

Please clarify the outcomes of the study. What tools or questionnaires were used to gather the data (e.g., blood tests, urine samples, lab reports, electronic files)?

Clearly define the primary and secondary outcomes, as this will help readers follow the results section.

In line 190, what is meant by "although other differences are similar in significance"? Please explain this clearly.

Results:

The results section should be rewritten using quantitative data instead of descriptive language.

Thank you for addressing these points. I look forward to the revised version of your manuscript.

Author Response

Reviewer 3

Dear Authors,

Thank you for your efforts in conducting this study. I found your work very interesting and would like to provide the following suggestions for improvement:

1. Please ensure the abstract is structured according to the appropriate study checklist.

Thank you, reviewer’s reminder. The summary has been rewritten and restructured accordingly.

2. The Methods and Results sections appear incomplete. Kindly revise and expand them to include all necessary details.

The Materials and Methods section have been revised and restructured in a standard format to include all necessary information.

3. Materials and Methods:

Please clarify the type of study. Is it retrospective or prospective? Is it a cohort or case-control study?

This is a case-control study.

4. How were the patients selected?

Subjects are selected according to the inclusion and exclusion criteria which are indictaed in the flow chart (Figure 1 in the manuscript). Prior to recruiting subjects in the study, all subjects received medical consultations in the Department of Endocrinology at the Republican Diagnostic Center in Astana/Kazakhstan. The diagnosis of PCOS patients is based on the presence of clinical symptoms and signs of hyperandrogenism and biochemical hyperandrogenism after exclusion of other causes of hyperandrogenemia according to the NIH and the Endocrine Society Clinical Practice Guidelines (Legro et al., 2013)

*Legro RS et al., 2103, doi: 10.1210/clinem/dgab248. PMID: 24151290.

5. When was the study conducted? What was the sample size formula? Why were the specified numbers of women included in the different groups?

Thank you very much for your questions.

This study was conducted from Sep 2013 to Dec 2014. The data was analyzed in 2023 – 2024.

For the sample size calculation, we used the specialized G*Power 3.1.9.7 software for the purpose of power calculation based on the input parameters, including available sample size, assumed effect size, desired significance level, but it does not provide a formula directly. In our study, we used an effect size of 0.8 (considered large), as PCOS is known to be associated with significant metabolic and hormonal alterations. We selected 25 participants for the PCOS group because of the prevalence of PCOS and the known challenges in recruiting patients with this condition. This is why the PCOS group had a smaller number of participants (n=25), while the non-PCOS obese group had more participants (n=67). The non-obese control group (n=46) was selected to provide a balanced comparison across the groups while maintaining statistical power for both comparisons and correlations. With this, the G*Power 3.1.9.7 software gave us a power of 0.92 for group comparisons and 0.99 for correlation analysis, which allowed us to reach enough statistical power for our analyses.

6. Were all lab tests and ultrasounds conducted by the same team or laboratory?

Yes, ultrasound investigations and all biochemical measurements were done at the Republican Diagnostic Center (RDC) in Astana/Kazakhstan by some of our team members.

7. Statistical Analysis:

Please provide details about the ROC analysis and how the area under the curve (AUC) was calculated. This should be included in the statistical analysis section.

Regarding the ROC analysis, we used it to assess the sensitivity and specificity of AMH as biomarker for PCOS diagnosis. The area under the curve (AUC) was calculated to show how well AMH can distinguish women with and without PCOS. The AUC value ranges from 0.5 (no discriminative power) to 1.0 (perfect discrimination). This analysis was performed using GraphPad Prism 10 software, as is now detailed in the Materials and Methods section (lines 193–197). We kindly ask the reviewer to specify which additional details are needed, so we can provide them accordingly.

We also included the following part of the ROC analysis in the Results section:

At the AMH threshold of 8.173 ng/mL, the sensitivity equals 90.00% (95% CI: 69.90% to 98.22%), and specificity is 97.26% (95% CI: 90.55% to 99.51%), indicating that at this level, 90% of women with PCOS can be correctly identified, while 97.26% of women without PCOS would be correctly classified if their AMH level is below 8.173 ng/mL. The likelihood ratio was 32.85, showing that women with AMH above this level are 32 times more likely to have PCOS.

8. Outcome:

Please clarify the outcomes of the study. What tools or questionnaires were used to gather the data (e.g., blood tests, urine samples, lab reports, electronic files)?

The main outcome of the study is that AMH and testosterone levels in obese PCOS patients were significantly higher than those in non-PCOS patients (regardless of obesity), and the serum AMH levels were positively correlated with testosterone levels.

Blood tests are used to gather the data using biochemical analyzers.

9. Clearly define the primary and secondary outcomes, as this will help readers follow the results section.

Primary outcome:  the serum AMH and testosterone levels in obese PCOS patients were significantly higher than those in non-PCOS patients (regardless of obesity), and AMH was also positively correlated with testosterone.

Secondary outcome: Serum AMH, testosterone, and LH were positively correlated in obese PCOS patients, while AMH and FSH were negatively correlated.

10. In line 190, what is meant by "although other differences are similar in significance"? Please explain this clearly.

Dear reviewer, thank you for noting this mistake, we deleted the phrase.

11. Results:

The results section should be rewritten using quantitative data

We really appreciate reviewer’s worthy suggestions. Now the Results section of the manuscript has been completely revised and re-written using quantitative data. Hopefully, it will meet the standard.

Thank you for addressing these points. I look forward to the revised version of your 

manuscript.

Round 2

Reviewer 1 Report

Comments and Suggestions for Authors

Authors have revised the manuscript based on the reviewers comments. 

Reviewer 3 Report

Comments and Suggestions for Authors

many thanks for the revision.